# When Good Guys Turn Bad: Bone Marrow’s and Hematopoietic Stem Cells’ Role in the Pathobiology of Diabetic Complications

**DOI:** 10.3390/ijms21113864

**Published:** 2020-05-29

**Authors:** Maria Cristina Vinci, Elisa Gambini, Beatrice Bassetti, Stefano Genovese, Giulio Pompilio

**Affiliations:** 1Unit of Vascular Biology and Regenerative Medicine, IRCCS Centro Cardiologico Monzino, I-20138- Milan, Italy; elisa.gambini@ccfm.it (E.G.); beatrice.bassetti@ccfm.it (B.B.); giulio.pompilio@ccfm.it (G.P.); 2Unit of Diabetes, Endocrine and Metabolic Diseases, IRCCS Centro Cardiologico Monzino, I-20138- Milan, Italy; stefano.genovese@ccfm.it

**Keywords:** diabetes, bone marrow, hematopoietic stem cells, inflammation, epigenetics

## Abstract

Diabetes strongly contributes to the development of cardiovascular disease, the leading cause of mortality and morbidity in these patients. It is widely accepted that hyperglycemia impairs hematopoietic stem/progenitor cell (HSPC) mobilization from the bone marrow (BM) by inducing stem cell niche dysfunction. Moreover, a recent study demonstrated that type 2 diabetic patients are characterized by significant depletion of circulating provascular progenitor cells and increased frequency of inflammatory cells. This unbalance, potentially responsible for the reduction of intrinsic vascular homeostatic capacity and for the establishment of a low-grade inflammatory status, suggests that bone BM-derived HSPCs are not only victims but also active perpetrators in diabetic complications. In this review, we will discuss the most recent literature on the molecular mechanisms underpinning hyperglycemia-mediated BM dysfunction and differentiation abnormality of HSPCs. Moreover, a section will be dedicated to the new glucose-lowering therapies that by specifically targeting the culprits may prevent or treat diabetic complications.

## 1. Introduction

Type 2 diabetes mellitus (T2DM) and also type 1 (T1DM) are associated with an increased risk of atherosclerotic cardiovascular disease (ASCVD) that represents the most prevalent cause of mortality and morbidity in diabetic patients [1,2]. Both T2DM and T1DM are characterized by hyperglycemia, and an enhanced inflammatory state that promotes and/or accelerates the molecular and cellular processes involved in the development of ASCVD in these patients [3]. The pauperization and the dysfunction induced by DM on bone marrow (BM)-derived circulating CD34^+^ hematopoietic stem/progenitor cells (HSPCs) are likely to play a key role in this scenario. In the last 15 years, the DM-associated dysfunctions of circulating progenitor cells have been one of the most revolutionary fields of study in diabetes. However, the preclinical and clinical data so far reported have mainly been interpreted in view of the putative cardiovascular protective and pro-angiogenic role of specific circulating progenitor subsets, overlooking the physiological function of HSPCs, which is to generate blood cell progenies in BM [4]. Only by combining all HSPC functions, which comprise both regenerative and immunological properties, we can comprehensively explain how DM-induced perturbation of these cells associates with sequelae of dramatic clinical events. Indeed, considering that HSPCs are common precursors of both pro-angiogenic and inflammatory cells, their dysfunction makes them both victims and perpetrators in DM and diabetic complications, respectively [5]. HSPC alterations are attributable to BM dysfunction, which is another target organ damaged by diabetic complications [6]. Speculatively, the defects of HSPCs could represent an epiphenomenon of BM impairment that become a central housekeeper of an organism’s health [7]. This review deals with the modulatory effects of the diabetic milieu on BM function, providing evidence of the mechanisms involved in BM dysfunction and alteration of differentiation programs of HSPCs. Finally, a section is dedicated to the recent advances in pharmacological management of diabetes, discussing the potential protective effects of new glucose-lowering agents on BM and HSPCs.

## 2. Endothelial Progenitor Cells: A Multifaceted Definition and a Unique Recognized Role in Diabetic Complications

CD34 is a transmembrane glycoprotein that is expressed by BM-derived HSPCs [8]. HSPCs are responsible for the generation of all adult blood cells, including specialized cells that control immune function, homeostasis balance, response to microorganisms, and inflammation [9].

Since 1968, HSPC transplantation therapy has been successfully used for the treatment of various malignant human blood disorders because of their ability to reconstitute all of the hemato-lymphoid system [10]. The vast majority of HSPCs resides in specialized niches within the BM, whereas a small number of stem and progenitor cells normally traffic throughout the body. However, the mechanisms underlying this trafficking as well as the physiological significance of their migration have remained relatively obscure. The current hypothesis is that the continual egression from the BM niche, extravasation into different tissues, and then their re-ingression into the BM, processes that are regulated by circadian rhythms [11], have the role of patrolling peripheral organs, with the aim of maintaining the immune responsiveness, tissue homeostasis, and regeneration. The continuous information exchange from the periphery to the BM, mediated by HSPCs and other factors, grants, beside a constant BM niche remodeling, a timely and efficient response to stressors. Indeed, different injury paradigms, such as hemorrhagic shock, inflammation, stroke, infection, and ischemia, are known to increase the circulating pool of HSPCs [12,13], even if the contribution of these cells to tissue repair and regeneration is still under investigation [14]. In 1997, Asahara and colleagues pioneered over 20 years of research in stem cell biology and regenerative medicine, which is still focused on the identification, isolation, and functional characterization of circulating BM-derived progenitors with vasculogenic properties [15]. Asahara, by identifying BM-derived circulating CD34- kinase insert domain receptor (KDR)-positive cells capable of incorporating into regenerating host vessels and contributing to vascular repair, challenged for the first time, the dogma that vasculogenesis is restricted to embryogenesis. These cells were named endothelial progenitor cells (EPCs). Since this pioneering work, nowadays, the search term “endothelial progenitor cells” in PubMed returns over 84,900 clinical, preclinical, and basic research articles, often controversial, in which these cells play the leading role (reviewed in [16,17,18]). The lack of consensus relies on two main reasons: The heterogeneous population of cells originated from the BM and the different methodological approaches used for their identification and characterization, such as surface marker antigens, cell culture, and clonal proliferation techniques [19,20,21]. So far, from a simplistic but usefulness point of view, two major EPC cell populations have been isolated by culture techniques from peripheral blood: The myeloid angiogenic cells (MACs, also called circulating angiogenic cells (CACs), pro-angiogenic hematopoietic cells (PACs) or “early EPCs”) and the endothelial colony-forming cells (ECFCs or outgrowth endothelial cells or “late EPCs) [18]. Both populations, although profoundly different in surface marker antigens and functions [21], share KDR, CD31, CD34, CD105, lectin, and acetylated low-density lipoprotein (acLDL) markers (Figure 1), and are thought to synergistically cooperate in the re-vascularization processes. MACs, without changing their lineage fate and integrating into the remodeled endothelium [22,23], support endothelial repair and regeneration through paracrine factors, whereas ECFCs, an endothelial cell type characterized by potent intrinsic clonal proliferative potential, structurally contribute to de novo blood vessel formation in vivo [24,25]. Considering tissue re-vascularization as a two-steps proces, it has been hypothesized that MACs are first actively attracted into the damaged tissue, where, by the release of soluble factors, ECFCs are recruited, hypothetically either from the circulation or local vessel wall. Next, MACs instruct ECFC proliferation and migration to the injury site to restore the endothelial integrity of the vascular wall [26,27]. The ECFCs, isolated from adult human peripheral blood and cord blood, display a clonal phenotype in vitro and cell surface markers undistinguishable from non-proliferative vascular endothelial cells. This rare cell population, whose BM origin is still largely debated [25,28,29], does not express progenitor cell markers and shows different functions and transcriptomic features compared with mature endothelial cells [30]. However, though ECFCs are widely accepted as putative EPCs in vitro, their effective existence and function in vivo are elusive.

### 2.1. Circulating HSPCs: Current Concept and Implication in Health and Disease

Despite the 20 years of research efforts dedicated to the identification of EPCs, the impossibility of sharply separating EPCs from hematopoietic cells, either progenitor or myeloid lineage-committed, is so far evident because of their overlapping phenotypes [31,32]. Today, many research groups are now inclined to identify circulating EPCs with the more generic CD34^+^/CD133^+^ HSPC population because it is the ancestor of all EPC phenotypes. The qualitative and quantitative depletion of circulating HSPCs, which encompasses progenitor subsets with multiple lineage commitment, has been shown to be independently associated with the risk of mortality in patients with coronary artery disease [33], renal disease [34], as well as with the severity of pathologies, such as Alzheimer’s [35] and cognitive impairment [36]. Such epidemiological observations derived from different cohorts of patients clearly suggest an important role of circulating HSPCs in the physiological mechanisms of endogenous tissue regeneration. This hypothesis is further supported by numerous preclinical and clinical studies showing the ability of these cells to repopulate non-hematopoietic tissues, including the kidney, heart, and lungs [37,38,39,40,41], as well as their efficacy, although controversial regarding the exact definition and function of such cells, in the treatment of cardiovascular disease, peripheral arterial disease, and critical limb ischemia in humans [42,43,44].

### 2.2. Circulating HSPCs and Diabetes: A Strong and Intertwined Relationship

Following EPCs’ identification, HSPCs are now considered integral components of the mechanisms involved in the maintenance of endothelial integrity, vascular homeostasis, and endogenous tissue regeneration in general. DM induces micro- and macro-angiopathies that over time lead to long-term damage and failure of various organ and systems. So far, numerous molecular mechanisms and pathways have been described to translate hyperglycemia, the hallmark of DM, into tissue damage [45]. However, dramatically, the same mechanisms seem to also be involved in impairment of the endogenous vascular reparative system, including stem cells. Considering that cell turnover is essential to face the physiological rate of cell senescence and death, it is easy to predict that in the subjects in whom intrinsic repair processes are defective, tissue damage can worsen. Under this prospective, the DM has been considered a disease of impaired damage control [46], which would explain the two- to three-fold increase of cardiovascular risk in this patient category. The discovery that diabetes affects stem/progenitor cell biology opened an important research field in diabetology, where stem cell dysfunction is at the center of the pathophysiological mechanisms of diabetic complications [47]. A consistent body of studies has documented the reduction and dysfunction in T2DM and T1DM of pro-angiogenic HSPCs (at the time still termed EPCs) [48,49,50,51,52]. These alterations, interpreted in view of a regenerative role, have been mechanistically considered to contribute to the onset and progression of vascular disease [53]. Importantly, a careful analysis of the same clinical studies reveals that pro-angiogenic HSPC alterations follow the natural history of diabetes, from its development to the later stage. This peculiarity suggests that the reduction of circulating pro-angiogenic HSPC numbers may also represent a biomarker of adverse cardiovascular events. Consistently, reduced levels of circulating pro-angiogenic HSPCs (EPCs) have already been associated with endothelial dysfunction, cardiovascular events, and with the pathogenesis of stroke in different cohorts of patients [54,55,56,57]. A recent meta-analysis by Rigato et al. showed that the numerical pauperization of circulating progenitors, in particular CD34^+^/CD133^+^ HSPCs, was associated with a 2-fold increased risk of future cardiovascular events and cardiovascular death in patients with suspected coronary artery disease, acute coronary syndrome, previous stroke, or in patients without acute events but with cardiovascular risk factors [58]. To note, the reduction of the same circulating cell population has recently been proven to be an efficient clinical-grade-independent biomarker in macro- and microvascular disease, capable of predicting long-term adverse cardiovascular outcomes in T2DM patients [59,60]. Though the molecular mechanisms mediating numerical and functional alteration of CD34^+^/CD133^+^ HSPCs in diabetes are still largely obscure (reviewed in [61]), BM is emerging as the original core of HSPC dysfunction.

To better clarify the mechanisms underpinning diabetic BM dysfunction, it is appropriate to introduce the complex cellular and molecular network regulating HSPC niche function.

## 3. Healthy BM

BM is the major hematopoietic organ and lymphoid tissue responsible for the production of erythrocytes, granulocytes, monocytes, lymphocytes, and platelets; moreover, it also represents the largest reserve of HSPCs in an adult organism [62]. Within the BM, HSPCs reside in a specialized microenvironment called the niche [63]. The BM niche, defined for the first time by Richard Shofield in 1978 [64], is an anatomical compartment composed by stromal cells, mesenchymal cells, osteoblasts, endothelial cells, reticular cells, and the extracellular matrix. The continuous molecular crosstalk between HSPCs and the other niche cellular constituents regulates HSC migration, quiescence, and differentiation [63,65,66].

So far, two distinct niches have been identified: Endosteal niche, lining the bone surface, and the vascular niche located around the sinusoidal endothelium. The two compartments are not separated but dynamically regulated and interrelated. Endosteal HSPCs are thought to mature during their relocation to the vascular niche; however, it is still unclear if endosteal and vascular HSPCs are mutually exchangeable [62]. The HSPC localization within the niches tightly correlates with the proliferative and maturation status of the cells. Indeed, HSPCs characterized by a high self-renewal capacity are localized near the endosteum, and as they mature, they move towards vascular sinusoids (vascular niche) [63,67], which play a pivotal role in progenitor cell egression from the BM to the circulation [68]. Osteoblasts of the endostal niche, which is further composed of a heterogeneous cell population, including fibroblasts and endothelial cells, regulate HSPC proliferation and quiescence [69]. There is evidence that osteoblast lineage cells are required to maintain the ‘endosteal stem cell niche’ as their expansion by genetic or pharmacologic means results in a concurrent expansion of HSPCs, whereas their ablation leads to HSPC loss [70]. In addition, osteoblasts also play a key role in HSPCs’ mobilization because they are a source of stromal cell-derived factor 1 (SDF-1/CXCL12 (CXC motif, ligand 12)), even if the interaction of this chemokine with its cognate receptor, CXCR4 (CXC motif, receptor 4) is not the only mechanism regulating HSPC trafficking in the BM. In this regard, numerous studies reported a very complex picture of the mechanisms regulating the homeostasis and mobilization of HSPCs that comprise not only endothelial cells and mesenchymal stromal progenitor cells but also mature hematopoietic neutrophils and macrophages. A recent review detailed the role that each of these cells of the HSPC niche play in contributing to mobilization [71]. Osteoblasts are also able to regulate HSPC survival, quiescence, and differentiation by producing factors, such as angiopoietin-1 [72] and thrombopoietin, capable of directly binding the HSPCs [73]. Stromal cells, on the other hand, are able to retain the HSPC in the BM by interacting between the vascular cell adhesion molecule 1 (VCAM-1) to the very late antigen-4 (VLA-4) antigen present on the HSPCs [74]. The maturation of the HSPCs is regulated by the Notch signal through interaction with osteoblasts [75].

The arterial vessels enter into the BM through the foramina nutricia and then divide into arterioles, capillaries, and sinusoids. Sinusoids, where the vascular niche is localized, consist of fenestrated endothelium and, to a lesser extent, pericytes, forming a permeable barrier for HSPC passage into the circulation [76]. While HSPCs located at the endosteum are more quiescent and have a greater self-renewal capacity due to a variety of cytokines, adhesion molecules, and hypoxia, HSPCs located close to the sinusoid endothelium (vascular niche) have reduced self-renewal capacity and cycle more rapidly due to higher oxygen levels. The cells of the vascular niche communicate with cells of the endosteal niche, and the subtle balance of factors from these sub-compartments governs the behavior of the HSPCs [77]. The vascular niche is involved in the proliferation and differentiation of HSPCs thanks to the presence of hormones, growth factors, oxygen, and nutrients from the peripheral circulation [78,79]. Moreover, the presence of different cell types, such as endothelial cells, mesenchymal cells, perivascular stromal cells (e.g., CXCL12-abundant reticular-CAR^+^, nestin^+^, and leptin^+^ receptor-Lepr+ cells), and Schwann cells [80,81,82], associated with HSPCs grant a fine-tuned regulation of stem cell quiescence and maintenance [83,84].

Overall, both endosteal and vascular niches support hematopoiesis and mobilization through paracrine signaling and physical interactions. Moreover, BM vascular cells direct the regular trafficking of HSPCs to the systemic circulation and back to the BM. Importantly, both niches are highly innervated [67]. Indeed, both niches are also finely regulated at the neuronal level by the sympathetic nervous system (SNS), which acts on perivascular cells through β3-adrenergic receptors. These signals that regulate CXCL12 expression in BM stromal cells allow HSPCs’ mobilization through circadian rhythms [85]. For example, nestin^+^ cells expressing the beta-adrenergic receptors, following noradrenergic stimuli, are able to downregulate the retention signals, allowing HSPC mobilization [78]. Similarly, granulocyte colony stimulating factor (G-CSF)-mediated HSPC mobilization is the result of SNS signaling on perivascular cells [86]. The SNS also acts directly on the HSPCs by the binding of adrenergic neurotransmitters (epinephrine and norepinephrine) and dopamine on β2-adrenergic receptors of HSPCs, which act as chemoattractants [87].

In homeostatic conditions, the number of HSPCs that physiologically leave the BM niche and reach the peripheral organs through the circulation to maintain tissue integrity is low. However, this number rapidly increases following tissue injury [62]. The mobilization of HSPC from the BM after injury is a delicate process that involves not only chemokine gradients and SNS activation [88,89] but also mediators, such as matrix metalloproteinases type 2 and 9; cathepsin K, which is secreted by osteoclasts; and elastases, produced by neutrophils [90]; all these proteases have the task of cutting the interactions between HSPCs and their niche microenvironment [62].

## 4. Diabetic BM

BM is now recognized as a target organ for chronic diabetic complications, in which defects, such as mobilopathy, neuropathy, and microangiopathy, are the main pathologic manifestations [76]. Profound alterations in gene expression and cytokine signaling induced by the diabetic microenvironment are the basis of changes in the BM cellular component, such as increased adipogenesis and decreased osteoblastogenesis [91].

Preclinical and clinical studies have implicated reactive oxygen species (ROS) overproduction, endoplasmic reticulum stress alterations, and advanced glycation end products (AGEs) in the abrogation of proangiogenic pathways involved in the restoration of perfusion and regeneration of ischemic tissue in DM. To these should also be added, for the sake of completeness, the marked impairment of BM-derived HSPC mobilization, recruitment, and pro-angiogenic differentiation [50,90,92,93,94,95]. These defects could depend on (i) reduced cell survival in the circulatory stream, (ii) altered homing out of the vessels, (iii) BM dysfunction with insufficient and less viable stem cell release, or (iv) a combination of all [7].

### 4.1. Mobilopathy

Mobilopathy is defined as the inefficient release of HSPCs from BM due to an altered gradient of chemokines both within the BM and between the BM and the periphery [67]. This functional defect can be partly explained by CXCL12/CXCR4 axis dysregulation [96]. Under homeostatic conditions, both the stem cell factor (SCF (also referred as KIT-ligand)/c-Kit (CD117) axis, and the binding between CXCL12, mainly produced by BM stromal cells, and CXCR4/CXCR7 receptors expressed on HSPCs keep the cells anchored to the niche. In response to peripheral ischemia, the decline of BM CXCL12 levels is the key signal of HSPC mobilization. In diabetes, however, CXCL12 in the BM does not decrease following ischemia, leading to an inefficient mobilization of HSPCs [97]. In this regard, Albiero and co-workers found an excess of proinflammatory M1 macrophages in the BM of a T1DM mice model. Moreover, they also found that oncostatin M (OSM) was the long-sought soluble factor released by macrophages that sustained CXCL12 expression by mesenchymal stem/stromal cells, which was no longer downregulated even after G-CSF stimulation. A few years later, the same authors also demonstrated that the oncostatin M (OSM)-p66Shc pathway mechanistically linked HSPC mobilopathy to excessive myelopoiesis [98,99].

In the ischemic tissue, the transcription factor hypoxia-inducible factor-1 (HIF-1) regulates CXCL12 gene expression in endothelial cells, which is directly proportional to reduced oxygen tension. HIF-1-induced CXCL12 expression increases the adhesion, migration, and homing of circulating CXCR4-positive progenitor cells to the ischemic site [100]. Diabetes is known to upregulate dipeptidyl peptidase-4 (DDP-4) expression, an enzyme involved in CXCL12 degradation, in the peripheral tissue. This DDP4-dependent reduction of CXCL12 levels results in the dampening of HSPC recruitment with a consequent angiogenesis deficit, as demonstrated in the heart of rats with chronic heart failure [101]. In the BM, on the other hand, DM inadequately upregulates DPP-4 on CD34^+^ cells, which is required for the mobilizing effect of G-CSF [102]. Nevertheless, DDP-4 inhibition was shown to promote HSPC mobilization by protecting CXCL12 from inactivation [103] and enzymatic degradation [104] in T2DM patients.

In addition, endothelial nitric oxide synthase/nitric oxide (eNOS/NO) pathway activation, fundamental for HSPC mobilization, is also compromised in diabetes, as demonstrated in a murine model of diabetes, where the eNOS enzyme was decoupled [105] and its phosphorylation rate reduced [106].

### 4.2. Inflammation

The increase of proinflammatory cytokines, such as interleukin-3 (IL-3), IL-10, IL-1β, tumor necrosis factor-α (TNF-α), and monocyte generation has been observed in the BM of a T1DM mice model [107]. Moreover, the downregulation of insulin-like growth factor 1, insulin-like growth factor-binding protein 5, osteoprotegerin, and vascular endothelial growth factor (VEGF) n the BM plasma of T1DM mice was associated with reduced Bmi1expression, a gene involved in senescence protection, and with an impaired HSPC repopulation of BM [108]. Hyperglycemia is also able to induce a small but specific BM cell subpopulation, identified by proinsulin production, that expresses TNF-α, a cytokine implicated in numerous diabetic complications, including diabetic peripheral neuropathy (DPN) [109]. This latter finding represents the first emerging evidence of active HSPC contribution to the pathogenesis of DPN.

### 4.3. Neuropathy

Sensory neuropathy is a common complication of DM characterized by damage of the sensory nerves. This causes an altered pain perception and impairs the activation of healing mechanisms after injury. Under physiological conditions, peripheral sensory neurons release, after damage, nociceptive factors, such as substance P (SP), which exert local and systemic actions, including recruitment to the site of damage of BM and circulating HSPCs expressing the neurokinin 1 receptor (NK1R), the main SP receptor. Recent studies demonstrated that sensory neuropathy can occur in the bone marrow of T2DM patients and mice. The rarefaction of nociceptive fibers in the diabetic BM was associated with an impairment in the neurokinin gradient between the BM, peripheral blood, and peripheral tissue, accompanied by a depressed recruitment of NK1R-HSPCs to the site of injury [110,111].

It is known that SNS is prominently involved in BM niche function [88] and stem cell trafficking is regulated by catecholaminergic neurotransmitters [87]; therefore, diabetic autonomic neuropathy (DAN) may also impact the BM. Albiero et al. demonstrated that BM denervation in an experimental model of diabetes (streptozotocin-induced and ob/ob mice) was mediated by p66Shc upregulation and that impaired mobilization relied on sirtuin 1 (Sirt1) dysregulation [112]. In this complex scenario, Lucas et al. showed that G-CSF is able to directly increase the sympathetic tone. Peripheral sympathetic nerve neurons express the G-CSF receptor and its stimulation with G-CSF reduced norepinephrine reuptake significantly, suggesting that G-CSF potentiates the sympathetic tone by increasing the norepinephrine availability. Their data suggest that the blockade of norepinephrine reuptake may be a novel therapeutic target to increase the stem cell yield in DAN patients [113]. Using a rat model of T2DM, Busik et al. demonstrated that BM neuropathy preceded the development of diabetic complications. They observed a reduced number of nerve endings in the BM of diabetic rats that coincided with a numerical increase of pro-angiogenic HSPCs within the BM and decrease of their circulating levels. In addition, denervation was also accompanied by a loss of HSPC circadian release, suggesting that inherent sympathetic denervation can alter the circadian peripheral clock. Overall, both DM-induced denervation and circadian clock impairment led to diminished reparative capacity, eventually resulting in the development of diabetic retinopathy [114].

### 4.4. Microangiopathy

Diabetes induces microvascular remodeling with a negative consequence for BM homeostasis. Oikawa et al. demonstrated that cultured endothelial cells from the BM of T1DM mice showed increased levels of oxidative stress, senescence, reduced migratory and network formation capacities, and increased permeability. BM endothelial cell dysfunction, in which numerous mechanisms, including increased expression and activity of Ras homolog family member A (RhoA) and reduced Akt phosphorylation/activity, are involved, is responsible for reduced BM perfusion and consequent HSPC loss. In addition, HSPC depletion induced by BM microangiopathy was associated with increased oxidative stress, DNA damage, and apoptosis of the stem cells [6,115].

### 4.5. Oxidative Stress

The reactive oxygen species (ROS) gradient is a fundamental signaling mechanism controlling functional niche compartmentalization of stem cells in the BM. The precious low replicating HSPCs, necessary for BM repopulation, reside in the “low ROS zones”, ideal for the maintenance of quiescence and retention, whereas the “high ROS zone” adjacent to the marrow vasculature has the function of facilitating stem cell maturation [6]. However, under pathologic conditions, such as diabetes, excessive ROS production [116] might jeopardize stem cell viability by promoting DNA damage and senescence [6,62]. Numerous mechanisms might account for increased oxidative stress in HSPCs, including BM hypoperfusion and hyperglycemia, both recognized as strong inducers of ROS generation by mitochondrial complex III, and exposure to ROS produced by other cell sources, such as endothelial and stromal cells of the niche [6,117,118]. In this context, Src homology/collagen (Shc) adaptor protein 66 (p66Shc), a gene that regulates the apoptotic responses to oxidative stress, plays a fundamental role. Di Stefano et al. showed that in vitro hyperglycemia-induced upregulation of p66Shc protein in HSPCs derived from mouse BM correlated with increased mitochondrial ROS production and a marked reduction of pro-angiogenic commitment of the cells. Conversely, p66Shc knockout BM-derived HSPCs were insensitive to high glucose exposure. Importantly, the high glucose resistance of p66Shc knockout BM-derived HSPCs was prevented by nitric oxide (NO) synthase inhibition, suggesting that the reduction of NO bioavailability induced by ROS was the underlying mechanism involved in pro-angiogenic HSPC dysfunction [119].

## 5. Evidence of Diabetes-Induced HSPC Programming and Pathobiological Implications

The BM harbors immature progenitor cell populations characterized by high plasticity and multiple cell lineages’ commitment, not limited to the hematopoietic system [120,121]. As previously reported, the stem cell niche has an important role in HSPC homeostasis [122,123]. This suggests that BM niche alterations induced by the diabetic milieu might have profound implications in HSPC biology, for example, by redirecting differentiation toward harmful cell populations that rather than protecting, promote the progression of diabetic complications. In this regard, numerous studies have reported the existence of BM-derived circulating progenitors with smooth muscle (smooth muscle progenitors, SMPs) [124,125] and calcifying phenotypes (osteoprogenitor cells, OPCs) [126,127], in DM patients. These cells, hypothetical “side products” of differentiation drift, witness the fact that diabetes not only reduces the number of vasculoprotective cells but also promotes the generation of cells with anti-angiogenic [128] and pro-fibrotic properties [129], with clear implications in diabetic micro- and macro-angiopathies’ development. In addition to mobilization defects of pro-angiogenic HSPCs and generation of aberrant progenitor cell populations, DM patients tend to have an elevation of peripheral inflammatory monocytes [130,131,132], an alteration in macrophage polarization [133], as well as increased levels of circulating inflammatory cytokines [134]. Terenzi and colleagues recently demonstrated the incontrovertible reduction of circulating proangiogenic progenitors and the increased frequency of circulating proinflammatory cells in TD2M patients [135]. Cumulatively, this unbalance, which culminates in a reduction of the intrinsic vascular homeostatic capacity and in the establishment of a low-grade inflammatory status [136], might be the main driver in atherosclerosis development and the increased risk of CVD in DM patients [137,138]. Loomans et al. for the first time hypothesized that hyperglycemia could modulate HSPC function. He demonstrated that hyperglycemia altered the differentiation fate of BM precursor cells, reducing the potential to generate vascular regenerative cells and favoring the development of proinflammatory cells [139]. After this publication, other studies based on the murine model of diabetes further substantiated and mechanistically detailed this finding [107,140,141].

Recent evidence supports the concept that epigenetic mechanisms may participate in the differentiation drift of HSPCs into more inflammatory cell populations within the BM. Epigenetic modifications have been documented in inflammation [142], and diabetic conditions are known to elicit epigenetic changes in a variety of cell types, including HSPCs [143,144]. In this regard, we recently demonstrated that a hyperglycemic condition induced inhibitory histone modifications and DNA methylation of the CXCR4 promoter in cord blood-derived CD34^+^ stem cells. These epigenetic alterations, also identified in BM-derived CD34^+^ stem cells of T2DM patients, were associated with a functional impairment of the CXCR4/CXCL12 axis [145]. This study, which demonstrates a direct impact of hyperglycemia exposure on the epigenetic make-up of stem cell promoters, further supports the hypothesis that epigenetic changes are likely additional mechanisms by which diabetes promotes the generation of inflammatory BM-derived monocytes and macrophages in diabetic patients, contributing to enhanced CVD. In a mouse model of insulin resistance, Gallagher and colleagues found that a repressive histone methylation mark, H3K27me3, is decreased at the promoter of the IL-12 gene in BM progenitors and this epigenetic signature is passed down to macrophages [146]. Similar findings, even if in different genes and experimental settings, have been described by other authors [147,148]. In particular, Yan et al. observed in T2DM mice DNA methyltransferase 1 (DNMT1)-dependent dysregulation of genes related to macrophage differentiation in HSPCs that was carried down through progenitor cells to terminally differentiated cells [148]. This study supports the concept that the gene expression pattern of terminally differentiated immune cells might be epigenetically “preprogrammed” by hyperglycemia at the HSPC level and may potentially be responsible for the prolonged and intrinsic inflammatory status of diabetics [138] (Figure 2).

Thus, epigenetic changes can be considered causative factors contributing to the differentiation drift of HSPCs in DM. Whether this alteration in DM is potentially reversible is not fully clear; however, there is growing body of evidence regarding the use of epigenetic drugs to restore/improve HSPC function. Multiple studies have reported that chemical modifiers of DNA demethylation, such as 5-azacytidine, or histone deaacetylase inhibitors, such as trichostatin A, valproic acid, or methyltransferase EZH2 inhibitors, enzymes that establish repressive H3K27me3 marks, have been shown to not only enhance both the plasticity and function of progenitor cells but also change the cell fate by chromatin remodeling [149,150]. Interestingly, the inhibition of bromodomain and extraterminal domain (BET) proteins by the food drug administration (FDA)-approved drug apabetalone (RVX-208) has been shown to modulate coagulation, vascular inflammation [151,152], and innate/adaptive immune responses, suggesting its potential use in the epigenetic treatment of chronic diabetic inflammation and vascular complications [153].

## 6. The Protective Role of Glucose-Lowering Drugs on the HSPC Level and Inflammatory Profile

Patients with T2DM are at high risk of developing ASCVD, the leading cause of morbidity and mortality in these patients [154]. Recent guidelines indicate that people with T2DM and established ASCVD, besides adequate glucose control, should be managed with glucose-lowering drugs proven to reduce major adverse cardiovascular events and/or cardiovascular mortality ([155,156], http://www.siditalia.it/clinica/standard-di-cura-amd-sid). Nevertheless, the mechanisms by which these drugs exert pleiotropic cardiovascular effects are not completely understood.

It is well known that HSPC dysfunction and systemic chronic inflammation are very common features in T2DM patients, and some reports have already shown that metformin can partially reverse these processes [157,158]. Accordingly, a number of studies have been performed (or are still ongoing) to evaluate whether therapeutic approaches with other glucose-lowering drugs, including peroxisome proliferator-activated receptor gamma (PPAR-γ) agonists, dipeptidyl peptidase 4 (DPP-4) inhibitors, glucagon-like peptide 1 (GLP-1) receptor agonists, and sodium-glucose transport protein 2 (SGLT2) inhibitors, may reduce or reverse these pathophysiologic factors in T2DM patients.

PPAR-γ agonists, also known as thiazolidinediones, were the first investigated. Several randomized and non-randomized clinical trials using PPAR-γ agonists in T2DM patients have shown positive results in terms of the endothelial function and inflammatory profile [159,160,161,162,163,164]. As an example, the largest randomized clinical trial by Esposito et al. [162], cumulatively enrolling 110 newly diagnosed T2DM patients, proved the superiority of pioglitazone versus metformin in improving the imbalance between endothelial damage (i.e., the endothelial microparticles expressing CD31) and the endothelial repair capacity (i.e., the endothelial progenitors expressing CD34/KDR). As for inflammation and oxidative stress, in a study on T2DM patients naïve to therapy, pioglitazone after 16 weeks significantly reduced CRP and E-selectin levels from baseline and in comparison to metformin, whereas no difference was observed for nitrotyrosine (a marker of oxidative stress) between the two treatments [165]. Interestingly, a prospective randomized controlled study that compared pioglitazone and rosiglitazone with the best standard of care showed that both treatments significantly decrease serum VEGF levels compared to controls (*p* = 0.00) and reduced thee serum TNF-α concentration, with a statistical significance for the pioglitazone group only (*p* = 0.01) [166]. Cumulatively, these results provide solid evidence for the anti-inflammatory and cardioprotective effect of PPAR-γ therapy and pioglitazone, which so far represents a valid therapeutic strategy in T2DM patients with established ASCVD ([155], http://www.siditalia.it/clinica/standard-di-cura-amd-sid).

As for DPP-4 inhibitors, sitagliptin and saxagliptin are the most widely investigated along with vildagliptin in this setting [104,167,168]. However, results are mixed. For example, the EDGE study (Effectiveness of Diabetes control with vildaGliptin and vildagliptin/mEtformin) revealed that 12 weeks of sitagliptin treatment increased circulating CD34^+^ cells (*p* = 0.03) but did not change inflammatory markers (i.e., high-sensitivity CRP and pentraxin-3) and oxidative stress markers (i.e., malondialdehyde-modified low-density lipoprotein and urine 8-hydroxy-2′-deoxyguanosine) [169]. Other investigators observed similar biological effects with saxagliptin for 12 weeks and vildagliptin for 12 months, respectively [167,168]. Conversely, other investigators did not find differences regarding both EPC functionality and the inflammatory profile in patients treated with different DPP-4 inhibitors [170,171,172,173]. In addition, a very recent network meta-analysis demonstrated the superiority of SGLT-2 inhibitors and GLP-1 agonists versus DPP-4 inhibitors in preventing cardiovascular events and mortality in this setting of patients [174].

Likewise, GLP-1 receptor agonists have been proposed for their protective role on vascular endothelium and the immune system [175,176,177]. Wei et al. [176] enrolled 31 newly diagnosed T2DM patients receiving lifestyle modifications plus incremental doses of exenatide (10 µg/day for 1 month and 20 µg/day for 2 months) or lifestyle modifications alone. This study showed that exenatide treatment significantly improved the endothelial function of coronary arteries by measuring the coronary flow velocity reserve (CFVR) and the system inflammation status by reducing the circulating levels of vascular adhesion molecules (i.e., soluble intercellular and vascular adhesion molecule-1). Similar results were also reported in other head-to-head comparison studies. For example, it was shown that exenatide and metformin treatments can equally improve endothelial dysfunction and inflammation [178], even in a pre-diabetes setting [179].

However, it is worth noting that overall, these trials did not adopt a placebo-controlled group. In addition to exenatide, the impact of liraglutide in T2DM patients is under investigation [180,181] but available data are still limited. A parallel-group study of liraglutide and glargine therapy showed a reduced deterioration of endothelial function for the group receiving liraglutide compared with controls as measured by flow-mediated dilation. However, this difference was not significant (5.7% to 5.4% and 6.7% to 5.7%, respectively) [181]. In a recent prospective randomized open-label trial, the administration of liraglutide and dulaglutide for 24 weeks produced the same antioxidant effect as demonstrated by improvements in the diacron-reactive oxygen metabolite and reactive hyperemia index [180]. However, this was an open-label study with a small sample size (n = 22). More definitive indications will arise from the ongoing clinical trials testing the role of semaglutide (NCT04126603) and liraglutide (NCT02686177) in regulating vascular integrity and angiogenesis.

More recently, investigators have focused on the novel drug class of SGLT-2 inhibitors [182,183]. Specifically, in the DEFENCE trial (dapagliflozin effectiveness on vascular endothelial function and glycemic control), Shigiyama et al. [184] compared the effect of dapaglifozin plus metformin and metformin alone in 80 early stage T2DM patients. At the end of the 16-week treatment period, the authors showed that the dapaglifozin add-on therapy compared to metformin-alone therapy significantly improves the flow-mediated dilation in those patients having HbA1c≥7.0% at baseline (1.05 ± 2.59 versus −0.94 ± 2.39; *p* < 0.05) and reduces urine 8-hydroxy-2′-deoxyguanosin, a clinical marker of oxidative stress (−0.6 ± 1.8 versus 1.1 ± 2.2; *p* < 0.001). In contrast, the EMBLEM trial (Effect of Empagliflozin on Endothelial Function in Cardiovascular High Risk Diabetes Mellitus) [185], in which a total of 117 adults with T2DM and established ASCVD were randomized to receive either empagliflozin 10 mg daily or placebo for 24 weeks, did not find differences in the primary endpoint (i.e., the change in the reactive hyperemia peripheral arterial tonometry index). Further insights will emerge from the ongoing randomized parallel-group trials. For example, the “Role of Canagliflozin on CD34^+^ Cells in Patients With Type 2 Diabetes” trial (NCT02964585) is currently recruiting patients with T2DM to study, as the primary endpoint, the gene expression and functional changes of CD34^+^ EPC. Other secondary endpoints, including serum endothelial inflammatory markers (hs-CRP, IL-6, and TNF-alpha), will also be investigated.

Interestingly, clinical trials are ongoing, and aim to evaluate the combination of glucose-lowering drugs possessing complementary and synergistic effects, such as saxagliptin plus dapaglifozin (NCT03660683) or empagliflozin plus liraglutide (NCT03878706).

## 7. Concluding Remarks

It is now clear that the BM is a target organ for chronic diabetic complications and HSPCs play an active role in the development of multiple organ vascular complications. The realization that DM may not only impair HSPC mobilization from the BM but also program the differentiation towards progenitors with defective vasculotrophic functions and excessive proinflammatory, pro-calcific, and fibrotic properties, with direct effects in the pathogenesis of diabetic complications, opens a new and exciting chapter of investigation in the diabetes field. From a therapeutic standpoint, the achievement of good glycemic control without glucose variability remains the best and primary way of preventing diabetic complications. However, a better understanding of the cellular and epigenetic mechanisms involved, as inciting events, in the differentiation drift of BM stem/progenitor cells towards deleterious cell populations in diabetes will help to identify new treatments to slow down or reverse diabetic complications.

## Figures and Tables

**Figure 1 ijms-21-03864-f001:**
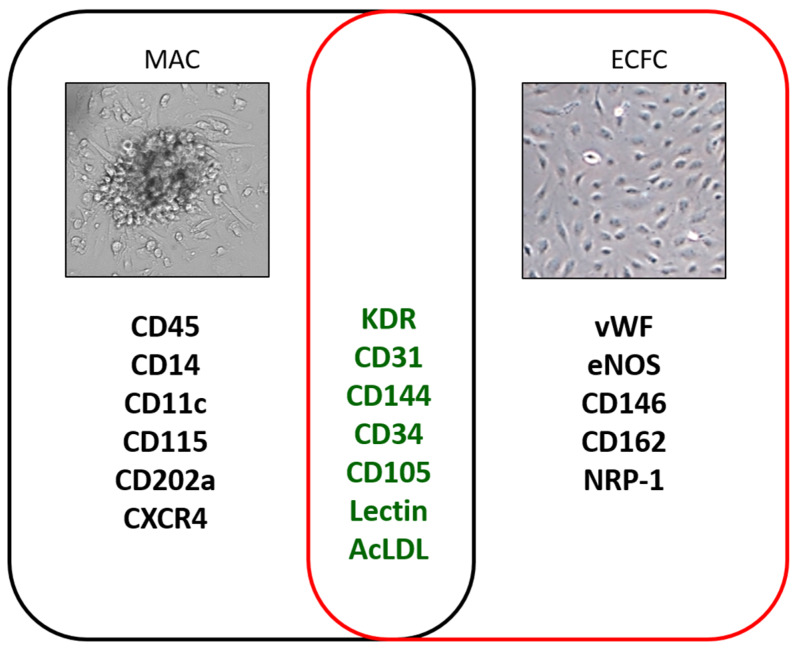
Cluster differentiation (CD) and other markers that identify myeloid angiogenic cells (MACs), black left circle, and endothelial colony forming cells (ECFCs), red right circle. The markers shared by both cell populations are in the center. Cluster domain (CD); CXC motif, receptor 4 (CXCR4); kinase insert domain receptor (KDR); acetylated low-density lipoprotein (acLDL); Von Willebrand factor (vWF); endothelial nitric oxide synthase (eNOS); Neuropilin-1 (NRP-1).

**Figure 2 ijms-21-03864-f002:**
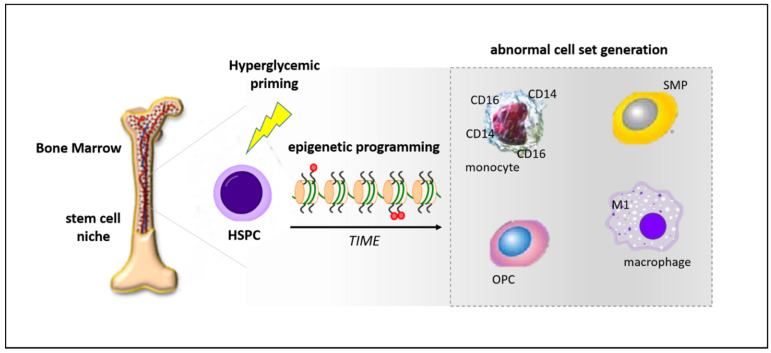
Schematic representation of hyperglycemia-induced programming of hematopoietc stem progenitor cells (HSPCs) at the bone marrow level. Diabetic milieu promotes epigenetic changes in HSPCs that result in the abnormal expansion of cells with inflammatory and pro-atherosclerotic features, such as intermediate monocytes (CD14+CD16+), M1 macrophages, osteoprogenitor cells (OPCs), and smooth muscle progenitors (SMPs).

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
