# Peer review of "When Good Guys Turn Bad: Bone Marrow’s and Hematopoietic Stem Cells’ Role in the Pathobiology of Diabetic Complications"

_ijms, 2020, doi:10.3390/ijms21113864_

Round 1
Reviewer 1 Report
From both clinical and animal work, a large body of studies have shown that hyperglycemia-induced dysfunction of bone marrow HSPCs, including reduced number and defective function of vascular-protective progenitor cells, contributes to micro- and macro- vascular complications of diabetes.
On the other hand, recent studies indicated that hyperglycemia-induced re-programming of HSPCs results in abnormal expansion of cells with inflammatory or pro-atherosclerotic features, which also potentially be responsible for inflammatory and ischemic status of diabetics.
In this manuscript, the authors comprehensively reviewed those works with important updates on studies of HSPCs differentiating toward to harmful cell populations, promoting complications of diabetes.
The manuscript is well written and organized. The work is scientifically sound and can be of great benefit to researchers in the field.
Author Response
Reviewer #1:
General comments:
From both clinical and animal work, a large body of studies have shown that hyperglycemia-induced dysfunction of bone marrow HSPCs, including reduced number and defective function of vascular-protective progenitor cells, contributes to micro- and macro- vascular complications of diabetes.
On the other hand, recent studies indicated that hyperglycemia-induced re-programming of HSPCs results in abnormal expansion of cells with inflammatory or pro-atherosclerotic features, which also potentially be responsible for inflammatory and ischemic status of diabetics.
In this manuscript, the authors comprehensively reviewed those works with important updates on studies of HSPCs differentiating toward to harmful cell populations, promoting complications of diabetes.
The manuscript is well written and organized. The work is scientifically sound and can be of great benefit to researchers in the field.
Response 1: We really appreciate your positive comments.

Reviewer 2 Report
Maria Cristina Vinci et al provided a comprehensive review of the roles of bone marrow and hematopoietic stem cell in the pathobiology of diabetic complications, including the molecular mechanisms underpinning hyperglycemia-mediated BM dysfunction and differentiation abnormality of HSPCs, as well as the new glucose lowering therapies as a prevention or treatment of diabetic complications. The authors provided an informative introduction and clear conclusion. The authors also point out that a better understanding of the cellular and epigenetic mechanisms that lead to differentiation drift of BM stem/progenitor cells may provide new treatment of diabetic complications. The authors are suggested to discuss these effects of small molecule that can affect the BM stem/progenitor cell differentiation and diabetic complications. One minor comment is that Figure 1 need to be adjusted to fit with the text.
Author Response
Reviewer #2:
General comments:
Maria Cristina Vinci et al provided a comprehensive review of the roles of bone marrow and hematopoietic stem cell in the pathobiology of diabetic complications, including the molecular mechanisms underpinning hyperglycemia-mediated BM dysfunction and differentiation abnormality of HSPCs, as well as the new glucose lowering therapies as a prevention or treatment of diabetic complications. The authors provided an informative introduction and clear conclusion. The authors also point out that a better understanding of the cellular and epigenetic mechanisms that lead to differentiation drift of BM stem/progenitor cells may provide new treatment of diabetic complications.
The authors are suggested to discuss these effects of small molecule that can affect the BM stem/progenitor cell differentiation and diabetic complications. One minor comment is that Figure 1 need to be adjusted to fit with the text.
Response 2: We thank you for your valuable comments to help improve our manuscript. As suggested, we briefly discussed in the paragraph “Evidence of diabetes-induced HSPC programming and pathobiological implications” the effects of some epigenetic drugs in stem cell differentiation and diabetic vascular complications (line 425-437).
Minor Comment: Based on your comment we adjusted the Figure 1 with the text (line 111-114).
